# CoMo: Learning Continuous Latent Motion from Internet Videos for Scalable Robot Learning

## Abstract

Unsupervised learning of latent motion from Internet videos is crucial for building generalist robots. However, existing discrete methods suffer from information loss and struggle with complex and fine-grained dynamics. We propose CoMo, which aims to learn more precise continuous latent motion from internet-scale videos. CoMo employs a early temporal feature difference mechanism to prevent shortcut learning and suppress static appearance noise. Furthermore, guided by the information bottleneck principle, we constrain the latent motion dimensionality to achieve a balance between retaining sufficient action-relevant information and minimizing the inclusion of action-irrelevant background noise. Additionally, we also introduce two effective metrics for more directly and affordably evaluating and analyzing motion and guiding motion learning methods development: **(i)** MSE of action prediction, and **(ii)** cosine similarity between past-to-current and future-to-current motion embeddings. Critically, CoMo exhibits strong zero-shot generalization, enabling it to generate effective pseudo actions for unseen videos. The shared continuous distribution of robot action and video latent motion also directly benefits the joint learning of unified policy. Extensive simulated and real-world experiments show that policies co-trained with CoMo pseudo actions achieve superior performance with both diffusion and autoregressive architectures.

## 1 Introduction

While large-scale Internet data has enabled impressive generalization in vision and language models (Achiam et al., 2023; Kirillov et al., 2023), robot learning remains limited by data scarcity, low diversity, and high heterogeneity. To enable effective scaling in robotics, a promising direction is to leverage abundant Internet video data. Consequently, a recent popular paradigm (Ye et al., 2024; Chen et al., 2024b;a; Bu et al., 2025; Bjorck et al., 2025) focuses on learning latent motion models from extensive video datasets. These approaches typically utilize an inverse dynamics encoder–forward dynamics decoder architecture within a self-supervised framework using video frame pairs. Specifically, they commonly employ the VQ-VAE objectives (Van Den Oord et al., 2017) to quantize learned motion representations, to generate pseudo action labels for the unlabeled video data.

However, real-world motion is inherently continuous, characterized by complex, fine-grained, and often uncertain dynamics. Representing such motion with discrete codebook inevitably leads to information loss and limits its generalization to novel motion patterns. Furthermore, the underlying vector quantization techniques often introduce training instability (Wang et al., 2025; Zhang et al., 2024) and face scalability challenges (Fan et al., 2024; Shi et al., 2024). Evidence from visual generation (Li et al., 2024b; Fan et al., 2024) and robot learning (Kim et al., 2025; Intelligence et al., 2025) also suggests that continuous representations can yield superior performance. This motivates a natural question: *Could we learn continuous motion representations from action-less videos?* Continuous latent motion enables more accurate representation of fine-grained inter-frame changes and inherently provides better consistency with the continuity of robot action.

A key reason previous works favored discrete latent motion, using a small codebook, was to inhibit the model collapse risk. When attempting to learn continuous motion directly, models are highly susceptible to 'shortcut learning'. Specifically, the encoder may capture excessive future frame background information, rather than focusing on the foreground motion, so that the decoder can

reconstruct pixel-level details. This might degenerate the model into a future-frame predictor, subverting its utility as an action prediction mechanism suitable for co-training unified robot policies.

To address this issue, our CoMo, introduces a early temporal feature difference strategy, inspired by temporal difference networks (Wang et al., 2021) in video understanding. Specifically, we remove the direct encoding of future frame and replace it with features difference between current and future frame before the encoder input, which serves to suppress static future frame background information while enhancing dynamic motion cues. This design significantly mitigates the aforementioned model collapse. Furthermore, to ensure learned continuous motion representations to serve as more effective pseudo actions for unified joint training with robot data, we carefully determine an appropriate embedding dimensionality guided by the information bottleneck principle (Pomerleau, 1991), seeking a balance between capturing sufficient motion detail and minimizing background noise.

Additionally, evaluating and analyzing the latent motion representation presents unique challenges. Our ultimate objective is to leverage latent motion as pseudo-labels for action-less video data to improve policy performance. However, policy success rate is affected by many external factors, is not a direct measure of latent motion, and often lacks stability and interpretability. Furthermore, training and evaluating policies is resource-intensive. To enable a more direct latent motion analysis and low-cost evaluation, we introduce two extra metrics: **(i) MSE of action prediction (MSE).** It assesses the action-relevant information within the latent motion. We train a MLP to predict ground-truth actions using these motion embeddings and report the MSE. A lower MSE signifies richer action-specific content. **(ii) Cosine similarity between Past-to-Current and Future-to-Current motions (S-PCFC).** It measures the similarity between motion from temporally symmetric segments relative to a central frame $z(o_{t-n}, o_t)$ (past-to-current) and $z(o_{t+n}, o_t)$ (future-to-current). Higher-dimensional latent motion, while capturing more richer action-relevant information, also inevitably introduces redundant, action-irrelevant details like future frame backgrounds, which harm pseudo-labeling. Statistically, lower S-PCFC under the same data indicates relatively less future frame background redundancy. Empirically, we find that combined MSE and S-PCFC effectively reflects policy success rate, with the best success rate when both are relatively low (a better trade-off).

Finally, CoMo can generate effective pseudo action labels for action-less video data. The consistent continuous distribution between robot action and video latent motion directly facilitates unified and joint policy learning, removing complex multi-stage pretraining and finetuning procedures ( (Ye et al., 2024)) or explicit two-stage motion-before-action pipelines ( (Bu et al., 2025)). Finally, extensive simulation and real-world experiments validate that CoMo provides more precise, effective pseudo action labels and achieves superior policy performance compared to those using discrete latent motion or naive continuous baseline. In summary, our main contributions are as follows.

- We propose CoMo, for unsupervised learning of more fine-grained continuous latent motion representation from Internet videos, featuring a simple yet effective early temporal difference mechanism to suppress static background noise and ensure more meaningful motion capture.

- We introduce MSE and S-PCFC metrics, enabling more direct, comprehensive, low-cost and stable evaluation for latent motion analysis and learning methods development.

- We generate more precise pseudo action labels using CoMo. The consistently continuous distributions of latent motion and robot action naturally facilitate the joint learning of unified policy.

- Extensive simulation and real-world experiments demonstrate the superior performance of policies trained using the CoMo pseudo labels, including both diffusion and autoregressive approaches.

## 2 RELATED WORK

**Learning from Internet Data for Robotic Manipulation.** Limited robot data restricts scalable policy learning. Incorporating large-scale internet videos can enhance generalization and efficiency (McCarthy et al., 2024). Prevailing methods predict signals from video data, either as implicit auxiliary tasks for improved learning (Cheang et al., 2024) or as explicit guidance for policy execution (Wen et al., 2023). These signals include future visual observations (Du et al., 2023; Cheang et al., 2024; Bharadhwaj et al., 2024a), affordances (Nasiriany et al., 2024; Bahl et al., 2023), object masks (Bharadhwaj et al., 2024b; Su et al., 2024), optical flow (Ko et al., 2023), human hand poses (Mendonca et al., 2023; Wang et al., 2023), and sparse point trajectories (Gu et al., 2024; Wen et al., 2023; Xu et al., 2024; Bharadhwaj et al., 2024c). A recent popular framework (Bruce et al., 2024; Schmidt & Jiang, 2023; Ye et al., 2024; Chen et al., 2024b;a; Bu et al., 2025; Bjorck et al., 2025) utilizes an inverse dynamics encoder–forward dynamics decoder architecture with unsupervised

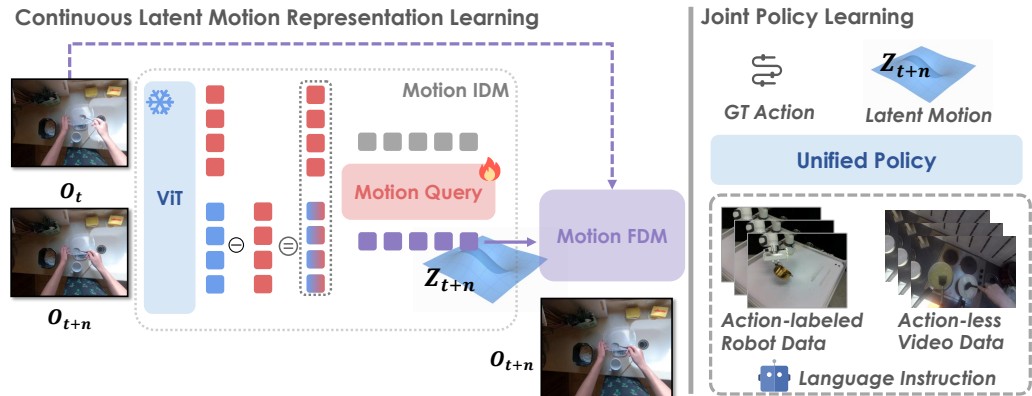

Figure 1: **CoMo framework.** We first self-supervisedly learn inter-frame latent motion from Internet videos. In the second stage, we directly utilize the trained IDM to extract pseudo labels for action-less videos, ensuring joint learning of continuous robot action and latent motion under a unified policy.

VQ-VAE (Van Den Oord et al., 2017) objectives to extract discrete latent motion from action-less videos. Additionally, concurrent works (Liang et al., 2025; Nikulin et al., 2025) introduce extra robot action supervision to train continuous motion encoders. In contrast, CoMo is purely self-supervised.

**Robotic Manipulation Policy Architecture.** Early works focused on state-based reinforcement learning (Andrychowicz et al., 2020; Joshi et al., 2020), while more recent methods leverage visual observations for imitation learning (Jang et al., 2022; Torabi et al., 2018; Fang et al., 2023; Yang et al., 2025). Many methods use generative probabilistic modeling (Brohan et al., 2022; Chi et al., 2023; Zhao et al., 2023) to capture the complex multi-modal action distribution. Among these, some methods (Brohan et al., 2023; O'Neill et al., 2024; Kim et al., 2024) adopt autoregressive-based policy architectures, which benefit from the generalization of pretrained VLMs but require discretizing robot actions. In contrast, others (Chi et al., 2023; Liu et al., 2024) use diffusion-based policy architecture to generate continuous robot action directly. Recent studies (Kim et al., 2025; Intelligence et al., 2025) have shown that continuous action representations enable finer-grained behavior modeling and often yield better performance. As a result, many advanced approaches (Wen et al., 2025; Li et al., 2024a; Black et al., 2024; Liu et al., 2025) combine autoregressive VLM backbones with diffusion-based action experts, thereby benefiting from both the strong generalization ability of pretrained VLMs and the expressive power of continuous action representations. On this basis, CoMo could leverage a unified policy architecture to jointly learn both continuous robot action and video latent motion.

## 3 METHOD

### 3.1 LEARNING CONTINUOUS LATENT MOTION WITH TEMPORAL FEATURE DIFFERENCE

We first describe our CoMo framework. CoMo adopts a inverse dynamics encoder–forward dynamics decoder paradigm, as illustrated on the left side of Fig. 1. Subsequently, we detail the technical aspects of our motion-enhanced inverse dynamics encoder and forward dynamics decoder respectively.

**Motion-Enhanced Inverse dynamics Model (ME-IDM).** Our ME-IDM aims to extract precise and background-irrelevant continuous motion information. Given a pair of the current frame $O_t$ and the future frame $O_{t+n}$, we use a shared MAE (He et al., 2022) pretrained ViT (Dosovitskiy et al., 2021) to extract their respective token-level features $F_t$ and $F_{t+n}$. Subsequently, to enhance motion cues and suppress static backgrounds, we perform a early temporal difference operation between $F_t$ and $F_{t+n}$ to obtain the token-level temporal feature differences $D_t$. To further alleviate the problem of shortcut learning when learning continuous motion representations, we explicitly remove the future frame features $F_{t+n}$ before the ecoder extracting motion embeddings. Specifically, we concatenate only the current frame features $F_t$ and the temporal feature differences $D_t$, resulting in the combined representation $[F_t, D_t]$. Following Moto-GPT (Chen et al., 2024b), we concatenate a set of learnable query embeddings with this token-level combined representation and perform full attention interaction within standard multi-layer Transformer layers (Motion Q-former). We then take the query features from the output of the Transformer layers as our motion representation $Z_t$.

**Forward dynamics Model (FDM).** Conditioned on the latent motion representation $Z_t$ obtained from the IDM and the current frame visual observation $O_t$, the forward dynamics decoder aims to

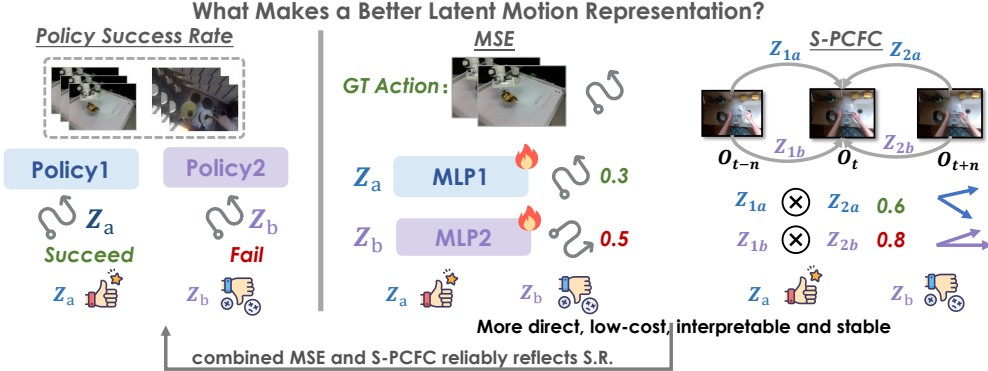

Figure 2: **The Metrics for evaluating and analyzing latent motion representations.** MSE and S-PCFC are used to analyze the representation of action-relevant information and future frame background information in latent motion, respectively. Better latent motion representations typically lead to higher policy success rate and lower MSE and S-PCFC.

reconstruct the the future frame observation $O_{t+n}$. Specifically, we first obtain a low-level patch-level embedding of $O_t$ using a linear patch embedding layer. Simultaneously, we further perform a pooling operation to compress the motion representation $\hat{Z}_t$, and then add the pooled motion feature to the low-level patch-level embedding of $O_t$, resulting in $E(O_t, Z_t)$. Subsequently, several Transformer layers process the combined features. Finally, the output features are processed using convolutional layers and a pixel shuffling operation to reconstruct the predicted future frame $\hat{O}_{t+n}$.

The CoMo framework is trained by jointly minimizing a weighted reconstruction loss and a perceptual loss to ensure both pixel-level accuracy and perceptual fidelity of the predicted future frames.

## 3.2 JOINT POLICY LEARNING

As illustrated on the right side of Fig. 1, we perform joint learning of action-less video data and continuous robot action data within a unified policy model. Specifically, given an action-labeled robot dataset $\mathcal{D}_R = \{\tau_1, ..., \tau_n\}$, where each $\tau_i$ represents a trajectory consisting of paired robot observations and actions, denoted as $\tau_i = [(o_0, a_0), ..., (o_T, a_T)]$, and a larger-scale action-less video data $\mathcal{D}_V$, we utilize the trained IDM to extract continuous latent motion embeddings for $\mathcal{D}_V$. As a result, each trajectory in $\mathcal{D}_V$ can be augmented as $[(o_0, z_0), ..., (o_T, z_T)]$, where $z_t$ denotes the latent motion inferred by the IDM at time step $t$. Since both $a$ and $z$ exhibit continuous data distribution, we can seamlessly perform joint imitation learning using the combined dataset $\mathcal{D}_R \cup \mathcal{D}_V$ within a unified generative policy model. The ability to leverage larger-scale dataset sources allows our CoMo to offer a scalable robot learning paradigm. In this work, we develop both a unified diffusion-based policy model and an autoregressive policy model.

## 3.3 MSE AND S-PCFC METRICS

Considering the policy success rate may be affected by many other factors, including high computational and data costs, we propose two extra metrics to more directly, stably and affordably evaluate and analyze latent motion representations, as shown in Fig. 2.

**MSE of action prediction (MSE).** Given an action-labeled robot dataset $\mathcal{D}_R$, we directly extract the latent motion embeddings at each timestep in an offline manner using the IDM trained in the first stage, resulting in a latent motion dataset $\mathcal{D}_Z$. We then train a extra MLP to map the latent motion embeddings to the corresponding ground-truth robot actions, and report the mean squared error of the prediction results in testing set. This process can be formally described as follows:

$$\hat{a}_t = \text{MLP}(z_t), \tag{1}$$

$$\text{MSE}(t) = \text{MSE}(a_t, \hat{a}_t). \tag{2}$$

**Cosine Similarity between Past-to-Current and Future-to-Current motions (S-PCFC).** While MSE reflects how well latent motion representations capture fine-grained inter-frame changes, it is only suitable for fair comparison when latent dimensionality is the same. From an information bottleneck (Pomerleau, 1991) perspective, latent motion should also minimize action-irrelevant noise,

such as static future frame background. However, MSE does not directly or fairly evaluate this, especially when comparing different latent dimensions. Therefore, we further propose S-PCFC. Specifically, given a sampled triplet of frames $\langle o_{t-n}, o_t, o_{t+n} \rangle$ from a short video clip, we extract the past-to-current motion $z(o_{t-n}, o_t)$ and the future-to-current motion $z(o_{t+n}, o_t)$. We then compute the cosine similarity between them, which can be formally defined as follows:

$$\text{S-PCFC}(t) = \frac{z(o_{t-n}, o_t)^{\top} z(o_{t+n}, o_t)}{\|z(o_{t-n}, o_t)\|_2 \, \|z(o_{t+n}, o_t)\|_2}. \tag{3}$$

Notably, S-PCFC assumes that latent motion inevitably contains action-irrelevant future frame background noise and that the sampled short video clips do not frequently exhibit periodic or repetitive motions. In our experiments, higher-dimensional latent motion representations inevitably introduce more background redundancy than low-dimensional robot action. These assumptions are satisfied, as S-PCFC is usually computed on approximately 1-second robot movement clips.

Overall, we demonstrate the effectiveness of both MSE and S-PCFC in evaluating and analyzing latent motion through qualitative analysis and quantitative results. Importantly, the combination of these metrics shows a strong correlation with downstream policy success rates. When both MSE and S-PCFC attain relatively low values (indicating a better trade-off between action-relevant information and future frame background redundancy), the downstream policy achieves the highest success rate.

## 4 EXPERIMENTS

We perform experiments using the LIBERO (Liu et al., 2023) and CLVIN (Mees et al., 2022) benchmarks and a Franka Emika Research 3 robot. Our experiments study the following questions:

**Q1:** Can self-supervised CoMo extract effective pseudo action labels for action-less video data and enable unified joint training with robot data to improve policy performance?

**Q2:** Do latent motion extracted by CoMo outperform other unsupervised predictive signals in videos?

**Q3:** Do continuous latent motion representations extracted by CoMo effectively mitigate model collapse and outperform discrete latent motion, naive continuous baseline, and other related methods?

**Q4:** Can continuous latent motion embedding dimensions extracted by CoMo be easily scaled?

**Q5:** Can MSE and S-PCFC effectively evaluate and analyze latent motion representations and reliably reflect downstream policy success rates?

### 4.1 SIMULATION EXPERIMENTS

#### 4.1.1 SIMULATION BENCHMARKS AND SETUPS

In this section, we describe our experiment setups for latent motion learning and evaluation, as well as unified policy learning and evaluation, on two simulation benchmarks. For more details, please refer to subsection A.5, A.6, and A.7.

**LIBERO.** The LIBERO (Liu et al., 2023) is divided into five categories: LIBERO-Spatial, LIBERO-Object, LIBERO-Goal, LIBERO-Long, and LIBERO-90. In our LIBERO experiments, we train two CoMo models for pseudo action labels extraction. We first use the entire in-domain LIBERO dataset to conduct ablation, which includes 13,000 videos from two camera views across 130 tasks. Then, to evaluate CoMo 's zero-shot cross-domain transfer capability, we then jointly train it on Internet videos by uniformly sampling 120,000 videos from SAM-V (Ravi et al., 2024), EgoVid (Wang et al., 2024), and Droid (Khazatsky et al., 2024), which cover in-the-wild, ego-centric human, and robot scenarios. For unified policy joint training, we adopt the data configuration of ATM (Wen et al., 2023), where each task is provided with only 10 robot trajectories and 40 video trajectories with pseudo action labels generated by CoMo. Regarding the policy architecture, following prior approaches (Chi et al., 2023; Liu et al., 2024), we implement a diffusion-based policy that conducts a joint denoising process within both the real robot action space and continuous latent motion space. For policy evaluation, we use the final epoch model and evaluate each task with 20 trials, repeating the evaluation three times to report the mean and standard deviation. In addition, to ensure a more robust and comprehensive ablation of the latent motion representation learning, we further report the MSE and S-PCFC results. These results are aggregated from 2,000 videos spanning 40 tasks across the four LIBERO suites.

Table 1: The ablation experiments results on the LIBERO benchmark.

| | Metric | O2-Fea | w/o. VQ | Pre-VQ | RGB-Diff | **Fea-Diff (CoMo)** |
|---|---|---|---|---|---|---|
| Spatial | Success Rate ↑ | 81.0±3.0 | 81.7±1.2 | 76.0±0.8 | 82.7±4.1 | 80.3±1.2 |
| | MSE ↓ | 1.208 | 1.189 | 3.055 | 0.891 | 0.881 |
| | S-PCFC ↓ | 1.000 | 0.988 | 0.821 | 0.786 | 0.892 |
| Object | Success Rate ↑ | 95.7±0.5 | 93.0±2.2 | 89.3±1.2 | 92.3±1.2 | 95.0±0.0 |
| | MSE ↓ | 0.896 | 0.865 | 2.363 | 0.604 | 0.662 |
| | S-PCFC ↓ | 1.000 | 0.992 | 0.810 | 0.810 | 0.902 |
| Goal | Success Rate ↑ | 78.3±1.7 | 89.0±2.4 | 74.7±0.5 | 85.0±2.4 | 85.0±2.2 |
| | MSE ↓ | 1.101 | 1.077 | 3.038 | 0.921 | 0.839 |
| | S-PCFC ↓ | 1.000 | 0.989 | 0.796 | 0.760 | 0.899 |
| Long | Success Rate ↑ | 47.7±3.3 | 47.0±1.6 | 54.3±0.5 | 59.0±5.1 | 63.0±1.6 |
| | MSE ↓ | 1.123 | 0.927 | 3.412 | 0.949 | 0.754 |
| | S-PCFC ↓ | 1.000 | 0.988 | 0.810 | 0.899 | 0.910 |
| Avg. | Success Rate ↑ | 75.7 | 77.7 | 73.6 | 79.8 | **80.8** |
| | MSE ↓ | 1.082 | 1.015 | 2.967 | 0.841 | **0.784** |
| | S-PCFC ↓ | 1.0 | 0.989 | **0.810** | 0.814 | 0.901 |

Table 2: The comparison results with other related methods on the LIBERO benchmark.

| | Spatial | Object | Goal | Long | Avg. |
|---|---|---|---|---|---|
| DP (5× data) | 92.0±3.3 | 96.3±0.9 | 93.0±2.2 | 75.3±2.0 | 89.2 |
| DP (1× data) | 72.3±2.5 | 82.3±0.5 | 70.3±0.5 | 56.7±3.0 | 70.4 |
| DP + ATM* (Wen et al., 2023) | 79.0±3.7 | 81.0±2.5 | 58.7±4.6 | 44.0±6.4 | 65.7 |
| DP + GR2-like (Cheang et al., 2024) | 81.0±3.0 | 95.7±0.5 | 78.3±1.7 | 47.7±3.3 | 75.7 |
| DP + GR00T (Bjorck et al., 2025) | 76.0±0.8 | 89.3±1.2 | 74.7±0.5 | 54.3±0.5 | 73.6 |
| DP + Dynamo (Cui et al., 2024) | 78.3±3.3 | 95.5±0.5 | 84.3±2.5 | 44.7±3.7 | 75.7 |
| DP + CoMo | 80.3±1.2 | 95.0±0.0 | 85.0±2.2 | 63.0±1.6 | **80.8** |

**CALVIN.** The CALVIN (Mees et al., 2022) benchmark is built upon the Franka robot and focuses on assessing long-horizon task completion. During each trial, the robot is required to sequentially complete five tasks. The benchmark consists of four different environments (A, B, C, D), allowing for robust evaluation of generalization capabilities. We conduct experiments under the most challenging ABC → D setup, training on environments A, B, and C, and evaluating on D. Following the setup of Moto-GPT (Chen et al., 2024b), we use all action-less videos from environments A, B, and C to train CoMo and 35% data (18k trajectory videos) with language annotations to conduct unified policy joint training. In terms of policy architecture, to enable joint prediction of continuous robot action and latent motion under a unified autoregressive-based policy framework, following (Chen et al., 2024b; Kim et al., 2025), we add two additional MLP networks after the final hidden states of the autoregressive decoder to jointly predict robot action and latent motion. As for policy evaluation, we select the last three epochs and report the mean and standard deviation of success rates.

### 4.1.2 SIMULATION EXPERIMENTS RESULTS AND ANALYSIS

To comprehensively explore different latent motion learning methods, we design and implement five primary ablation variants: **O2-Fea** (using the future frame ViT features as latent motion), **w/o. VQ** (continuous latent motion by naively removing vector quantization of prior works Ye et al. (2024); Chen et al. (2024b)), **VQ** (discrete latent motion by applying vector quantization, following prior works Ye et al. (2024); Chen et al. (2024b)), **RGB-Diff** (removing vector quantization and replacing the original future frame ViT features with the ViT features of the RGB difference between frames as input, to suppress background noise), and **Fea-Diff (Our CoMo)** (removing vector quantization and replacing the original future frame ViT features with the frame-wise difference of ViT features as input, to suppress background noise). As for VQ, considering the assumption of continuous data distribution in diffusion (Ho et al., 2020; Lipman et al., 2022; Liu, 2022), and following GR00T (Bjorck et al., 2025), we utilize pre-quantized embeddings as the latent motion in diffusion-based policy, referred to as **Pre-VQ**. Although these embeddings are continuous, the commitment loss imposed during training encourages them to approximate a discrete distribution. To ensure fair comparison, all the above variants use nearly identical architectures and parameter counts. The main differences are

Table 3: The experiment results on the CALVIN benchmark.

| | 1 | 2 | 3 | 4 | 5 | Avg. Len. |
|---|---|---|---|---|---|---|
| w/o. Motion | 0.814±0.016 | 0.586±0.029 | 0.412±0.030 | 0.294±0.033 | 0.200±0.024 | 2.306±0.119 |
| VQ | 0.824±0.024 | 0.619±0.031 | 0.446±0.025 | 0.341±0.020 | 0.247±0.019 | 2.477±0.084 |
| w/o. VQ | 0.828±0.013 | 0.637±0.013 | 0.481±0.016 | 0.387±0.020 | 0.299±0.024 | 2.632±0.148 |
| RGB-Diff | 0.815±0.022 | 0.628±0.037 | 0.475±0.025 | 0.375±0.031 | 0.285±0.034 | 2.577±0.184 |
| Fea-Diff (CoMo) | 0.854±0.021 | 0.684±0.039 | 0.538±0.044 | 0.438±0.041 | 0.334±0.043 | **2.848±0.129** |

whether vector quantization is applied and what input is used for the IDM. For our experiments, both VQ and Pre-VQ employed a codebook with a vocabulary size of 128.

In Tab. 1, we present ablation studies of the above variants on LIBERO, reporting S.R., MSE, and S-PCFC. In Tab. 2, we further compare the policy performance of CoMo with other related methods on LIBERO. In Tab. 3, we report ablation results of the core design of CoMo on CALVIN. In Tab. 4, we provide additional ablation results of the core design of CoMo on LIBERO under out-of-domain data settings. Overall, these results address the aforementioned questions and make following findings:

*Result 1:* The results in Tab. 2 indicate that incorporating video data with CoMo pseudo labels into the diffusion-based policy can increase the success rate on LIBERO from 70.4% to 80.8%. Meanwhile, the results in Tab. 3 show that CoMo also improves CLVIN's performance from 2.306 to 2.848.

*Finding 1:* CoMo provides effective pseudo labels for action-less video data. The ability of incorporating much richer data sources leads to a more powerful policy performance, making CoMo a more scalable and data-efficient learning paradigm, including diffusion and autoregressive architecture.

*Result 2:* In Tab. 2, achieves the best policy performance. GR2-like and GR00T correspond to O2-Fea and Pre-VQ, respectively. Dynamo utilizes a covariance regularization loss to suppress shortcut learning. Overall, we incorporate the core designs of the compared methods into our unified diffusion policy. The use of the same framework and data ensures the fairness of the comparison.

*Finding 2:* In the comparison of different predictive signals from video data to improve robot policy, the CoMo latent motion outperforms future frame visual features, 2D point trajectory, pre-quantized latent motion of GR00T, and regularized latent motion of dynamo.

*Result 3:* We emphasize the following comparison results: (i) Policy Success Rate: As shown in Tab. 1, for diffusion-based policy architectures, CoMo achieves an average success rate of 80.8%, consistently outperforming all other methods. In particular, it surpasses Pre-VQ by a notable margin of 7.2 (increasing from 73.6% to 80.8%). Furthermore, the results in Tab. 4 demonstrate that this conclusion remains valid when CoMo is trained on larger-scale, out-of-domain Internet videos, achieving an even higher policy success rate (from 80.8% to 81.8%). Meanwhile, as shown in Tab. 3, CoMo also attains the highest performance within the autoregressive-based policy architecture. Compared to using discrete latent motion, the results improve from 2.477 to 2.848. (ii) MSE: The results in Tab. 1 also indicate that CoMo achieves the best performance on the MSE metric, with a substantial margin over Pre-VQ (0.784 vs. 2.967). Similarly, the results in Tab. 4 further demonstrate that the conclusion still hold when training is extended to out-of-domain Internet video data. (iii) S-PCFC: For S-PCFC, the results in Tab. 1 and Tab. 4 jointly show that discrete latent motion achieves the lowest value, regardless of whether in-domain data or out-of-domain internet video data are used for training. For the other approaches, naively removing VQ only achieves a result close to 1.0 (0.989). When employing temporal difference mechanism for static background suppression, RGB difference performs better on in-domain data (0.814 vs. 0.901), whereas feature difference yields better results on out-of-domain internet video data (0.873 vs. 0.851).

*Finding 3:* Based on the above results, we have the following findings and analysis: (i) Although extracting discrete latent motion and explicitly imposing vector quantization constraints can effectively mitigate the shortcut learning problem, this approach leads to significant information loss. As a result, the extracted motion struggle to capture the robot's fine-grained movements (the highest MSE). As shown in Fig. 4, discrete latent motion may only capture the general movement direction. On the other hand, continuous latent motion and robot action share a consistent continuous distribution, and this consistency is beneficial for the joint learning of a unified policy. (ii) Simply removing the vector quantization leads to a severe shortcut learning problem, where the model tends to collapse by directly learning substantial future frame background noise as continuous latent motion. In

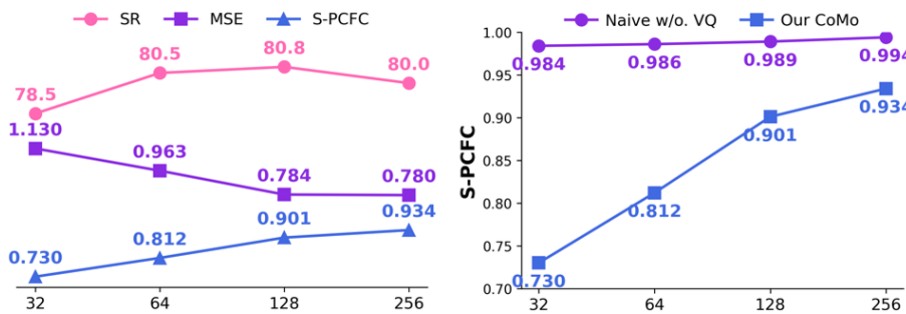

Figure 3: **The latent motion embedding dimension scaling results. (Left)** Our CoMo achieves the highest success rate when the motion dimension is 128 (the better trade-off between MSE and S-PCFC). **(Right)** Our CoMo obtains a lower S-PCFC as the motion dimension decreases, whereas simply removing vector quantization does not (with severe shortcut learning problem).

contrast, employing temporal difference mechanisms can effectively alleviate this issue (Both MSE and S-PCFC effectively reduce). As shown in Fig. 3(right), for CoMo, gradually reducing the motion embedding dimensionality effectively lowers the S-PCFC, whereas the naive w/o VQ baseline does not achieve the same effect. The visualization in Fig. 4 also confirms this: compared to naive continuous baseline, the predicted future frame of our CoMo include much less of the prompt video's background noise. Regarding the choice between RGB difference and feature difference, the former amplifies low-level motion signals more effectively, but may disrupt more abstract and complex motion information. Consequently, feature difference demonstrates greater scalability when applied to larger-scale internet video data (Both MSE and S-PCFC are lower). **(iii)** CoMo trained on large-scale internet videos exhibits strong generalization capabilities, enabling direct zero-shot transfer to robotic scenarios for generating continuous pseudo action labels without the need for fine-tuning. Moreover, the more complex motion patterns in real-world data can further mitigate the shortcut learning problem (The S-PCFC can be reduced from 0.901 to 0.851), thereby enhancing policy performance (from 80.8% to 81.8%).

*Result 4:* As shown in Fig. 3(left), increasing the dimensionality of the latent motion embedding initially leads to improved policy success rates, but further scaling results in a decline. This trend diverges from observations reported in prior work (Ye et al., 2024). Notably, during the incremental increase of the motion dimensionality, S-PCFC exhibits a marked improvement (from 0.730 to 0.940), whereas the MSE metric remains relatively stable in the later stages. For example, when the motion dimension increases from 128 to 256, the MSE only slightly decreases from 0.784 to 0.780.

Table 4: The LIBERO experiment results of CoMo trained on larger-scale Internet videos without any fine-tuning.

|  | S.R. ↑ | MSE ↓ | S-PCFC ↓ |
|---|---|---|---|
| w/o. VQ | 77.2 | 1.368 | 0.895 |
| Pre-VQ | 78.7 | 2.226 | **0.405** |
| RGB-Diff | 80.6 | 1.291 | 0.873 |
| Fea-Diff (Ours) | **81.8** | **1.177** | 0.851 |

*Finding 4:* The above results indicate that continuous latent motion representations cannot be simply scaled up. From the perspective of Information Bottleneck theory (Pomerleau, 1991), increasing the dimensionality of motion embeddings may encourage the model to extract more static appearance information. With a fixed policy model capacity, this is unfavorable for the joint prediction of low-dimension robot action and latent motion. The MSE and S-PCFC metrics variations further corroborate this point: although scaling up the motion dimension might capture more comprehensive motion information, the simultaneous introduction of additional background noise also impedes accurate action regression. Therefore, in this work, we fix the dimensionality of the latent motion representation to 128. This ensures that the model achieves a better balance between learning more informative motion cues and reducing static motion-agnostic background information.

*Finding 5:* In Tab. 1, Tab. 4, and Fig. 3, we present S.R., MSE, and S-PCFC on LIBERO across various methods, pre-training data, and latent dimensions. Overall, reducing MSE or S-PCFC alone does not guarantee better policy performance. The best policy performance is achieved when both are relatively low, indicating a better trade-off. Although a strict quantitative relationship cannot be established, the combination of MSE and S-PCFC reliably reflects downstream policy success. For

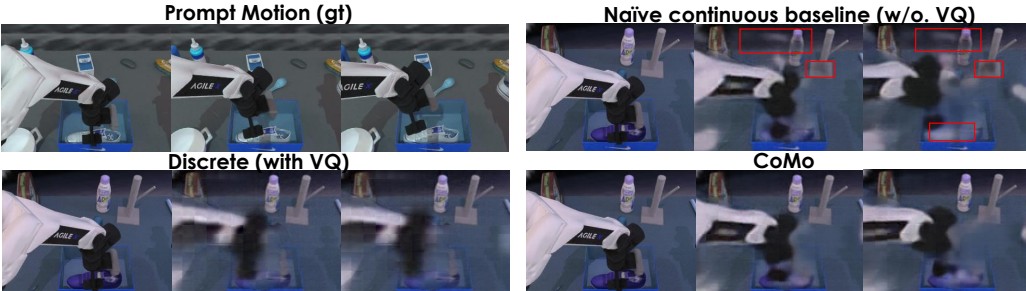

Figure 4: **The FDM future frame prediction visualization.** Given three frames from a prompt video clip, we extract the latent motion from the first two frames and from the first and last frames, respectively. We then render a new environment for the first frame, and use these two sets of latent motions to predict the subsequent two frames in the new environment via FDM. The red rectangles indicate that the naïve continuous baseline incorporates a significant amount of background information from the prompt video. In addition, compared to the other two baselines, CoMo provides finer-grained latent motion that more closely matches the trajectory in the prompt video.

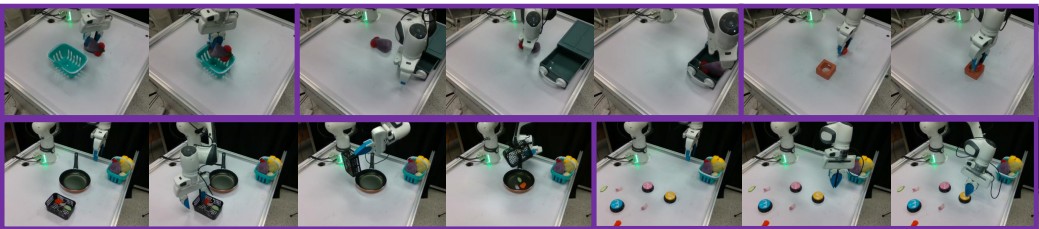

Figure 5: **The Real-world task illustrations.**

example, compared to the naïve continuous baseline, our CoMo achieves lower MSE, lower S-PCFC, and higher policy success rate.

## 4.2 REAL-WORLD EXPERIMENTS

**Real-World Setups.** In real-world experiments, we aim to validate whether CoMo trained on Internet videos, can directly extract latent motion from human videos to serve as effective pseudo action labels. Specifically, we utilize a Franka robot to execute five tasks: picking a toy and placing it into

Table 5: Real-world experiments.

|  | Pick | Open | Insert | Push | Pour |
|---|---|---|---|---|---|
| w/o. human videos | 60.0 | 15.0 | 20.0 | **80.0** | 20.0 |
| Pre-VQ | 65.0 | 25.0 | 15.0 | **80.0** | 25.0 |
| CoMo | **75.0** | **30.0** | **25.0** | 80.0 | **35.0** |

a basket, opening a drawer and placing a toy into it, inserting a toy into a container, pressing the button of the corresponding color, and pouring the vegetables from the basket into the pot, as shown in Fig. 5. For each task, we utilize 20 teleoperated trajectories and 20 human manipulation videos, where we construct pseudo action labels via our CoMo training on Internet videos for the latter. We then train a unified diffusion-based policy using both datasets and evaluate each task with 20 rollouts, and the results are presented in Tab. 5.

**Real-World Results and Analysis.** The results demonstrate the effectiveness of CoMo latent motion in extracting pseudo-action labels from human demonstration videos. Consistent with the conclusions from simulation experiments, CoMo latent motion achieves better policy than Pre-VQ (GR00T), which can be attributed to its continuous distribution that matches real robot action, more precise motion capture, and reduced action-irrelevant background noise.

## 5 CONCLUSIONS

We presented CoMo, a framework for self-supervised learning of continuous latent motion from Internet videos. By employing the straightforward temporal feature differences to replace future frame features before the encoder input, CoMo effectively mitigates shortcut learning issues, thus enabling more effective and precise pseudo action labels for action-less video data. This seamlessly facilitates the joint learning of continuous robot action and latent motion within a unified policy. Furthermore, we propose MSE and S-PCFC for a more direct and low-cost evaluation and analysis of latent motion. Extensive experiments results demonstrate the effectiveness of our CoMo.

ETHIC STATEMENT

This research was conducted in full compliance with the ICLR Code of Ethics. Our approach is based on publicly available datasets and benchmarks, with a primary focus on advancing algorithmic techniques. We have carefully considered all potential risks and broader societal impacts and have concluded that this work does not present any notable ethical concerns, nor does it involve the use of sensitive data or high-risk applications.

REPODUCIBILITY STATEMENT

To guarantee the reproducibility of our results, we offer the following resources:

- **Implementation details**: We provide detailed descriptions of our CoMo model and training procedures, as well as policy training and evaluation details, in subsections A.5,A.6, andA.7.
- **Code and models**: Our source code and trained models will be released upon paper acceptance to facilitate reproducibility and encourage further research.

We believe these resources establish a solid foundation for the research community to replicate and extend our work.

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

# A APPENDIX

## A.1 LIMITATIONS AND DISCUSSION

Despite extensive simulation and real-world experiments validating CoMo's effectiveness and generalizability in extracting pseudo action labels for action-less videos, and showing its superiority over discrete latent motion and naive continuous variants, some challenges and future works remain. As indicated by the results in Tab. 2 (89.2 vs. 80.8), there is still a gap between the latent motion and actual robot action. A potential future work involves incorporating extra temporal supervision to obtain more temporally sensitive, skill-centric motion representations, which we leave for future works. Furthermore, we hope our proposed low-cost and stable MSE and S-PCFC metrics will further facilitate the exploration and research of better latent motion learning methods.

## A.2 THE MSE METRIC FOR ABSOLUTE JOINT STATES

In the main text, the MSE metric is computed and reported using relative end-effector poses as the action space. In this section, we further supplement the analysis by reporting MSE results with absolute joint states as the action space. As presented in Tab. 6, the results demonstrate that the early temporal difference mechanism in our CoMo latent motion remains effective even when evaluated in the absolute action space. This indicates that when the input to the IDM consists of current frame features and inter-frame feature differences, it is still able to capture the spatial information of the foreground in future frames.

Table 6: The MSE metric for absolute joint states.

| MSE | Spatial | Object | Goal | Long | Avg. |
|---|---|---|---|---|---|
| Pre-VQ | 0.872 | 0.991 | 0.562 | 1.831 | 1.064 |
| w/o. VQ | 0.156 | 0.163 | 0.085 | 0.187 | 0.148 |
| CoMo | **0.122** | **0.121** | **0.074** | **0.173** | **0.126** |

## A.3 THE DATA SCALABILITY OF CoMo

Table 7: The data scalability of CoMo.

| Out-of-domain video data | S.R. | MSE | S-PCFC |
|---|---|---|---|
| 30,000 | 78.3 | 1.253 | 0.868 |
| 120,000 | **81.8** | **1.177** | **0.851** |

In the main text, we demonstrate the generalization capability of CoMo when trained with Internet-scale out-of-domain video data. In Tab. 7, we also report results obtained by training with only one-quarter of the data, which further verifies the data scalability of CoMo.

## A.4 REAL-WORLD EXPERIMENTS DETAILS

In this section, we detail the specifics of our real-world experiments. Specifically, our experiments setup is illustrated in Fig. 6, which comprises a single Franka Emika Research 3 robot arm, equipped with a UMI Chi et al. (2024) gripper, and utilizes a statically positioned RealSense D435 camera (with a resolution of 640×480 pixels) from a third-person view to acquire real-time RGB visual observations. Following available code[1], we employ a 3D mouse for teleoperation data collection. The robot system operates at 20 Hz (moderately reduced from the native 100 Hz control frequency to balance training efficiency and motion continuity), with actions defined as relative end-effector pose changes in SE(3) space (3D position change + quaternion orientation change).

For the five tasks we evaluated—picking a toy and placing it into a basket, opening a drawer and placing a toy into it, inserting a toy into a container, pressing the button of the corresponding color, and pouring the vegetables from the basket into the pot—they respectively require the robot arm to

---

[1] https://github.com/UT-Austin-RPL/deoxys_control

perform basic pick-and-place, long-horizon open-pick-place, fine-grained pick-insert, instruction following, and pick-pour capability. During evaluation, the initial pose of the robot arm was set to a fixed home position. The initial poses of the objects to be interacted with were significantly varied. A special case is the long-horizon open-pick-place task, where adhesive was applied to the bottom of the drawer to mitigate significant sliding during opening and closing. Consequently, in this task, the placement pose of the drawer was slightly perturbed, within a range of approximately 8 cm in the lateral and longitudinal directions.

For the policy of our real-world experiments, we adopt a diffusion-based policy architecture. Specific training and architecture details can be found in Section A.6. Finally, we jointly train the policy using collected robot data and human video data labeled with our Internet trained CoMo (CoMo-Internet).

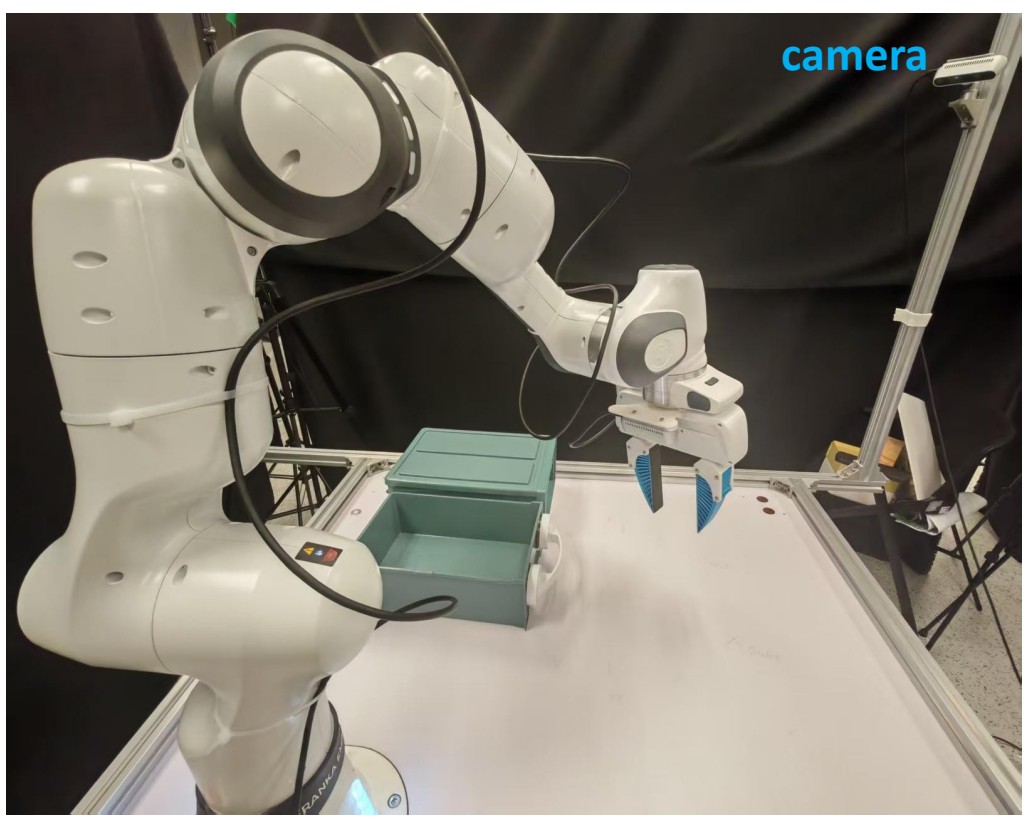

Figure 6: The real-world Franka robot arm experiments hardware platform.

### A.5 CoMo Details

In this section, we describe the specifics of our CoMo. Utilizing different training data, we train three versions of CoMo: CoMo-Internet (using sampled SAM-V Ravi et al. (2024), EgoVid Wang et al. (2024) and Droid Khazatsky et al. (2024)), CoMo-LIBERO Liu et al. (2023) (using entire LIBERO dataset), and CoMo-CALVIN Mees et al. (2022) (using entire CALVIN data of environments A, B, and C) to conduct ablation studies across different simulated environments and real-world scenario, and to validate the zero-shot cross-domain transfer capability of CoMo. We train them using largely the same training hyperparameters as detailed in Tab. 8. Notably, for a fairer comparison with the discrete baseline in Moto-GPT Chen et al. (2024b), we train CoMo-CALVIN for a longer duration. For different datasets, we set different frame intervals to ensure that they maintain approximately similar FPS.

Following Moto-GPT Chen et al. (2024b), CoMo employs a set of learnable motion queries to capture the latent motion representations between two frames. Specifically, our CoMo comprises a Motion-Enhanced Inverse Dynamics Encoder (ME-IDM) and a Forward Dynamics Decoder model (FDM). To mitigate shortcut learning problem and model collapse, in contrast to prior works, CoMo early remove future frame visual features before the IDM encoder input, and only utilize both the

feature differences between the current and future frames and the current frame visual features as inputs to suppress static backgrounds and enhance motion.

Finally, in Tab. 8, we further report extra architectural details of our CoMo. Notably, our CoMo-CALVIN employs a higher motion embedding dimensionality, which is used for a fairer comparison with the discrete baseline in Moto-GPT Chen et al. (2024b).

Table 8: The training and architectural hyperparameters for our CoMo learning.

| Hyperparameter | Value |
|---|---|
| *CoMo training* | |
| Optimizer | AdamW Kingma (2014) |
| Base learning rate | 0.0001 |
| Optimizer momentum | $\beta_1, \beta_2 = 0.9, 0.99$ |
| Effective batch size | 256 |
| Total training steps | 50,000 (150,000 on CALVIN) |
| Frame interval on LIBERO Liu et al. (2023) | 10 |
| Frame interval on CALVIN Mees et al. (2022) | 5 |
| Frame interval on SAM-V Ravi et al. (2024) | 10 |
| Frame interval on EgoVid Wang et al. (2024) | 10 |
| Frame interval on Droid Khazatsky et al. (2024) | 20 |
| *Motion-Enhanced Inverse dynamics Model* | |
| Feature extractor | MAE He et al. (2022) ViT-L |
| Codebook size of discrete baseline | 128 |
| Number of motion queries | 8 |
| Latent motion embedding dimensionality | 16 (32 on CALVIN) |
| #layers | 4 |
| #MHSA heads | 12 |
| Hidden dim | 768 |
| *Forward dynamics Model* | |
| #layers | 12 |
| #MHSA heads | 12 |
| Hidden dim | 768 |

## A.6 DIFFUSION-BASED POLICY DETAILS

In this section, we detail our unified diffusion-based policy. We primarily implement the diffusion-based policy for the LIBERO Liu et al. (2023) simulation and real-world experiments. Specifically, we jointly learn the unified policy from video data with continuous pseudo action labels constructed using CoMo, and continuous robot action data.

In Tab. 9, we report the training and architectural details of our diffusion-based policy. Specifically, we employ BERT Devlin et al. (2019) and ViT Dosovitskiy et al. (2021) to extract language instructions and visual observations features, respectively. Following RDT-1B Liu et al. (2024), we utilize a more scalable DiT Peebles & Xie (2023) block as the backbone. The extracted language and visual features are incorporated as conditioning through cross-attention layers within the DiT block. To perform joint learning of action-less video data and robot data within a unified policy model, we construct two sets of MLP networks to map continuous latent motion and robot actions into a shared embedding space, and back to their respective original spaces. In the training phase, we adopt the DDPM scheduler with a glide cosine scheduling scheme (specifically, the squaredcos cap v2 variant) across a diffusion process of 1000 steps. Conversely, for inference, we leverage the DPM-Solver++ Lu et al. (2022) in conjunction with an analogous glide cosine scheduler, albeit with a substantially reduced sampling budget of 5 steps. Finally, to capture the temporal dependencies of actions and ensure real-time dynamic adaptability during policy execution, we set an action / motion chunk size of 8 in both the training and inference phases.

Table 9: The training and architectural hyperparameters for our diffusion-based policy learning.

| Hyperparameter | Value |
|---|---|
| *Diffusion-based policy training* | |
| Optimizer | AdamW Kingma (2014) |
| Base learning rate | 0.0005 |
| Effective batch size | 256 |
| Total training epochs | 100 |
| *Diffusion-based policy architecture* | |
| Vision feature extractor | DINOv2 Oquab et al. (2023) ViT-B Dosovitskiy et al. (2021) |
| Language feature extractor | BERT Devlin et al. (2019) |
| #layers | 12 |
| #MHSA heads | 16 |
| Hidden dim | 768 |
| Action / motion chunk size | 8 |
| Action projector | (7, 768) |
| Latent motion projector | (128, 768) |
| Action head | (768, 7) |
| Latent motion head | (768, 128) |
| *Noise scheduler* | |
| Type | DDPM Ho et al. (2020) |
| Prediction type | sample |
| Training step number | 1000 |
| Sampling step number | 5 |
| Solver | DPM-Solver++ Lu et al. (2022) |

## A.7 AUTOREGRESSIVE-BASED POLICY DETAILS

In this section, we detail the specifics of our autoregressive-based policy, as hown in Tab. 10. We primarily implement this policy for the CALVIN Mees et al. (2022) simulation environment experiments. Specifically, we employ T5 Raffel et al. (2020) and ViT Dosovitskiy et al. (2021) to extract token-level textual and visual features, respectively. Following Chen et al. (2024b); Kim et al. (2025), we adopt a GPT-style Radford et al. (2018) autoregressive backbone and append two additional MLP networks at the output layer to predict continuous robot actions and latent motion separately. Specifically, for motion prediction, we autoregressively predict latent motion with a chunk size of 2. For action prediction, we parallelly decode actions with a chunk size of 5 based on a set of learnable action query tokens. Furthermore, to ensure a fair comparison with the discrete baseline in Moto-GPT Chen et al. (2024b), we first perform a round of pre-training using action-less video data before conducting joint training on robot action data and action-less video data.

## A.8 STATEMENT ON LLM USAGE

Our use of Large Language Models (LLMs) in the preparation of this manuscript was limited strictly to polish writing.

Table 10: The training and architectural hyperparameters for our autoregressive-based policy learning.

| Hyperparameter | Value |
|---|---|
| *Autoregressive-based policy training* | |
| Optimizer | AdamW Kingma (2014) |
| Base learning rate | 0.0005 |
| weight decay | 0.0001 |
| Effective batch size | 512 |
| Total training epochs | 20 |
| *Autoregressive-based policy architecture* | |
| Vision feature extractor | MAE He et al. (2022) ViT-B Dosovitskiy et al. (2021) |
| Language feature extractor | T5 Raffel et al. (2020) |
| #layers | 12 |
| #MHSA heads | 12 |
| Hidden dim | 768 |
| Action chunk size | 5 |
| Motion chunk size | 2 |

