# OpenReview forum: "CoMo: Learning Continuous Latent Motion from Internet Videos for Scalable Robot Learning"
_ICLR.cc/2026/Conference — ICLR 2026 Conference Withdrawn Submission_

### Official Review · Reviewer_hPiJ · 2025-10-29

**Soundness:** 3
**Presentation:** 1
**Contribution:** 2
**Rating:** 2
**Confidence:** 4

**Summary:**

The paper introduces CoMo, a framework for learning continuous latent motion from videos in a self-supervised manner, designed to address the limitations of prior *discrete latent motion models* (e.g, VQ-based methods). The main contributions are:
1. A temporal feature difference mechanism that suppresses static background information and prevents shortcut learning.
2. Two evaluation metrics for low-cost, interpretable analysis of latent motion.
3. A policy learning framework that jointly trains on robot trajectories and videos using the continuous latent motions as pseudo-labels.

Experiments on LIBERO, CALVIN, and real-world Franka manipulation tasks show competitive success rates compared to discrete latent motion baselines and competing methods. The approach claims better generalization, scalability, and robustness.

**Strengths:**

1. Clear motivation: continuous latent motion is more naturally aligned with robotic tasks than discrete latent motion, which can be restrictive and less representative of real-world dynamics.

2. Experiments: the proposed approach shows good empirical performance and the paper is complemented with real-world experiments.

**Weaknesses:**

The paper is not well written and clearly presented, which makes it difficult to verify and fully agree with many of the authors’ claims. In several places, the paper gives the impression of overselling its contributions, with statements such as “Unsupervised learning of latent motion from Internet videos is crucial for building generalist robots” that are strong, arguable, and insufficiently supported. Below I provide a detailed review:

**Clarity**:
The figures and explanations are sometimes confusing.
1. There is a mismatch between the way the modules are introduced in the method section and how they are depicted in Fig. 1.
2. Fig. 2 is particularly unclear; it is not explained in the text and does not seem to convey meaningful information, making it easy to skip without understanding its relevance.
3. The paper contains small but noticeable typos (e.g., “ecoder” in line 155, “CLVIN” in line 237).
4. More generally, many questions arise while reading the paper due to insufficient explanations (see questions below).

**Overstated novelty**
1. The claimed novelty largely relies on replacing a discrete latent space with a continuous one for modeling motion tasks. While the intuition is reasonable, the experimental evidence provided is not strong enough to justify the claim that continuous latent motion is fundamentally better. The bold claims in the Introduction are not sufficiently backed by the results.
2. The method can be reduced to a relatively simple combination of removing future frame features, temporal differencing, and bottleneck dimension control. This makes the contribution appear incremental relative to prior work on latent action/motion models.
3. The proposed evaluation metrics (MSE and S-PCFC) are heuristic and lack theoretical grounding. Their correlation with policy performance is only weakly demonstrated and seems noisy.

**Experiments**
1. Given the ambitious motivation laid out in the Introduction, I would have expected much stronger improvements over discrete latent space baselines. Instead, the reported gains are only marginal in most cases.

**Questions:**

1. ViT pretraining: How is the Vision Transformer pretrained? On which dataset(s), and under what training objective?

2. Figure 1: What does the label “Language instruction” represent in the diagram? Is natural language used explicitly, or is this symbolic?

3. Unified policy: Does the unified policy output both actions and latent motion simultaneously, or are these separate branches? Please clarify.

4. Figure 2: Could you provide a more detailed explanation of what Fig. 2 illustrates? Currently, it is difficult to extract meaningful insights.

5. Joint policy learning: In Section 3.2, why is it valid to combine $D_R$ and $D_V$? How is the action space defined when mixing robot trajectories with video-based pseudo-labels?

6. Metrics usage: Are the introduced metrics (MSE and S-PCFC) used solely as evaluation tools, or do they also play a role during training?

7. Table 1 categories: Could you provide more detail about the experimental categories in Table 1? What differentiates them and how should they be interpreted?

8. Evaluation protocol: For Tables 1 and 2, how is evaluation carried out? Specifically, how many random seeds are used, and how many episodes per seed are run to compute the reported results?

---

### Official Review · Reviewer_DiLH · 2025-11-01

**Soundness:** 2
**Presentation:** 2
**Contribution:** 2
**Rating:** 4
**Confidence:** 4

**Summary:**

The paper proposes a pipeline for learning continuous latent motion representations from raw videos using an IDM/FDM-style setup like LAPA [1], Moto-GPT [2] with feature differencing and by dropping raw future features to reduce shortcut learning. The learned motion latent is then used with a policy head (diffusion or autoregressive) and evaluated on LIBERO/CALVIN plus a small real-robot demo. The paper introduce two proxies for “latent quality” (a regression MSE from latent to action and a cosine similarity between Past-to-Current and Future-to-Current motions) and report some gains over several internal baselines.

While the approach is simple and well-motivated to reduce shortcut learning, the paper’s central claim of capturing fine, motion-centric structure is weakened by the coarse frame interval chosen for latent learning (Table 8). Using such a large stride effectively can effectively smoothen out high-frequency motions (e.g., contact onsets, fingertip adjustments, slip), making it unlikely that the learned latent encodes the very dynamics the paper highlights.

[1] Latent Action Pretraining from Videos

[2] Moto: Latent Motion Token as the Bridging Language for Learning Robot Manipulation from Videos

**Strengths:**

- Dropping raw future features and using feature differences is a reasonable way to reduce background leakage and can be implemented easily to existing latent action models.
- Introducing proxy metrics for latent quality is a constructive step toward more principled analysis of motion representations, even if the proposed metrics need further validation.
- A good ablations study where multiple input/representation variants are compared within the same overall framework.

**Weaknesses:**

- The core mechanism, feature differencing to induce motion-centric latents, has in fact been a part of LAPA, where feature differencing is applied before quantization (see their NSVQ module code). The current work appears to shift where differencing is applied (earlier in the pipeline) and additionally drops raw future features to limit shortcuts. Without a direct ablation that compares stage placement (early vs. late differencing) and with/without future-feature removal under matched network capacity, it is difficult to attribute the reported gains to a genuinely new idea rather than to incremental design tuning of an established recipe.
- VQ baseline likely under-regularized due to a large codebook. The ablation uses a codebook of 128, whereas LAPA uses a significantly smaller codebook of 8 while training on a larger dataset. A larger codebook weakens the information bottleneck, making background leakage more likely. As a result, the current comparison does not isolate whether the observed gap stems from “continuous vs. discrete” or simply from an high capacity VQ configuration.
- The statement "GR2-like and GR00T correspond to O2-Fea and PreVQ" is overly reductive. While GR00T indeed includes a PreVQ-style component, it also combines multiple design choices (architectural, training, data/augmentation, scaling) that materially contribute to its performance; reducing it to PreVQ mischaracterizes the method. Likewise, GR-2’s core claim is that visual future prediction can strongly benefit policy learning; treating GR-2 as an O2-Fea proxy downplays this. In fact, recent evidence (e.g., UVA [1]) further supports that future prediction does indeed benefit policy learning. The paper should more carefully contextualize these works and avoid implying that their contributions are captured by the O2-Fea / PreVQ ablations.

[1] Unified Video Action Model

**Questions:**

- How does the proposed early differencing compare to late differencing before quantization (as in LAPA)? How much gain comes purely from stage placement?
- What happens with smaller codebook size for VQ? Do conclusions hold when VQ is appropriately bottlenecked?
- What is the rationale for the coarse frame interval used in latent learning? Is the method still effective when reducing the frame interval, or does it need additional changes (model, losses, etc.) to tackle it?

---

### Official Review · Reviewer_QTUt · 2025-11-01

**Soundness:** 2
**Presentation:** 2
**Contribution:** 2
**Rating:** 2
**Confidence:** 4

**Summary:**

This paper proposes a method (CoMo) to learn continuous latent motion representations from action-free videos to improve robot policy learning. Unlike prior approaches that rely on quantization to suppress noise in latent actions, CoMo replaces the discrete codebook with a continuous latent space, aiming to capture richer motion representations. By combining latent motion with robot actions as mixed pseudo-action labels, CoMo achieves moderate improvements in policy success rates across simulated and real-world manipulation tasks. In addition to success rate, the paper evaluates latent motion quality using MSE and S-PCFC, but the correlation between S-PCFC and actual latent quality lacks clear empirical evidence.

**Strengths:**

Learning robot policies from video data is an important and challenging direction. Representing motion through latent actions is a promising approach, and exploring continuous latent spaces provides a new perspective. The method is validated across several environments, including LIBERO, CALVIN, and real-world robot setups, demonstrating basic applicability.

**Weaknesses:**

1. **Limited Contribution**
- The main innovation is the removal of quantization in latent-action learning, compressing information directly in a continuous space. While interesting, this change is modest and not theoretically or empirically justified.
- The design of joint policy learning largely follows GR00T-N1, with minimal architectural distinction.
- The use of action probing (MSE) as a latent-quality metric is not novel, and the interpretation of S-PCFC as correlated with representation quality is ambiguous. Given motion continuity, a lower similarity between temporally adjacent latent actions could stem from egocentric camera jitter or background variations, rather than meaningful improvement. Similarly, higher success rates might result from noise-induced robustness rather than better representations. Both the authors’ interpretation and alternative explanations can be plausible but lack evidence.

2. **Insufficient Experimental Support**
- Experimental details are unclear. For example, why was the codebook size of 128 chosen for comparison with variants with quantizer (considering LAPA uses 8192)? This drastically limits latent capacity, yet no rationale is provided.
- Data composition is opaque. While the paper claims to use 120K Internet videos, the actual number used for training is unspecified. Additionally, an analysis of the correlation between the performance and the action-free video quantity is lacked. what about  the ratio between robot demonstrations and human videos?
- S-PCFC interpretation is unsupported. Additional analyses (e.g., controlled perturbations or latent–visual correlation tests) are needed to verify its validity as a representation-quality metric.
- Missing key ablations: The paper seems to lack an ablation that removes only the temporal-difference operation. Directly computing ViT feature differences may be unreliable in egocentric viewpoints, where motion is dominated by camera movement rather than object dynamics. Without such ablations or discussion, the contribution of this operation remains unclear.

3. **Concern on the Scalability and Extension**
- A major motivation of using Internet videos is to scale up policy learning for better generalization, yet the current dataset (120K clips) is relatively small. Whether CoMo can scale to significantly larger datasets remains unknown.
- The purpose of quantization in prior works is to filter redundant information for scalability; it is unclear whether CoMo’s continuous latent space can achieve the same robustness when scaling.
- The video selection criteria are not specified—e.g., what fraction are egocentric videos, and can arbitrary Internet videos be used?
- In the LIBERO environment, many works (e.g., GR00T, Pi0) already achieve >90% success rate, likely due to stronger VLM backbones. Can CoMo integrate such stronger perception models to further improve performance?

**Questions:**

Please refer to the weakness part. Additionally：

1.	In Table 2, what does the abbreviation DP stand for—Diffusion Policy? This should be clarified.
2.	There are several typos (e.g., “CLVIN” should be “CALVIN”).
3.	In the RGB-Diff variant, is the same ViT backbone used? Was it frozen or tuned on downstream data, or does a pretrained ViT naturally handle RGB difference inputs?
4.	The temporal-difference idea resembles optical-flow modeling. Have the authors considered using explicit optical-flow features to replace or enhance this module?

---

### Official Review · Reviewer_BoA9 · 2025-11-03

**Soundness:** 2
**Presentation:** 3
**Contribution:** 1
**Rating:** 2
**Confidence:** 4

**Summary:**

This work improves latent action model by turning the discrete bottleneck into continuous one. The key innovation to achieve this is feed the latent action model with the difference between two frames, instead of individual frames. Experimental results cover both simulation and real-world benchmark.

**Strengths:**

* Easy to follow. The writing is great.

* Comprehensive experiments on both simulation and real-world scenarios.

* In general, combining action-labeled data and unlabeled videos makes sense, and I expect it will give benefits.

**Weaknesses:**

* A main concern is that I think the solution here to bridge the gap between continuous and discrete latent motion is quite like a trick, rather than a method. In principle, from an information perspective, giving (O_{t} - O_{t+n}, O_t) is equivalent to (O_{t+n}, O_t), which is learnable within the neural network. Therefore, the results are not convincing to me. Maybe due to some settings like hyperparameters, or training strategies. I can't figure out why it works. And I think maybe a KL penalty on the latent distribution may also works, but it's still a trick.

* Second, the motivation to transform discrete latent to continuous one is still not clear to me. I agree with that the discrete solution is motivated by preventing 'shortcut' from O_{t+n}, but I can't see any evidence to show that it is necessary to using continuous representation. This is not a reconstruction task, so information loss on details may not the key.

* I would suggest to firstly introduce which data did you use for the two models. How to train the proposed latent action model and policy models. It is very hard to find key messages in the long text.

* The proposed metrics, which I think the motivation is great because a challenge for latent action model is how to measure the performance. However, these two metrics are designed for continuous representation, which go against the discrete one in nature. Is it fair?

**Questions:**

I post the questions in weakness.

---

### Note · Authors · 2025-11-12

**Comment:**

We would like to thank all reviewers for their dedicated work. Following the submission, we have continuously improved the quality of our work. In future submissions, we will optimize our method and strengthen our experimental results and analysis.

**Withdrawal Confirmation:**

I have read and agree with the venue's withdrawal policy on behalf of myself and my co-authors.